# MONGE-AMPÈRE FLOW FOR GENERATIVE MODELING

## ABSTRACT

We present a deep generative model, named Monge-Ampère flow, which builds on continuous-time gradient flow arising from the Monge-Ampère equation in optimal transport theory. The generative map from the latent space to the data space follows a dynamical system, where a learnable potential function guides a compressible fluid to flow towards the target density distribution. Training of the model amounts to solving an optimal control problem. The Monge-Ampère flow has tractable likelihoods and supports efficient sampling and inference. One can easily impose symmetry constraints in the generative model by designing suitable scalar potential functions. We apply the approach to unsupervised density estimation of the MNIST dataset and variational calculation of the two-dimensional Ising model at the critical point. This approach brings insights and techniques from Monge-Ampère equation, optimal transport, and fluid dynamics into reversible flow-based generative models.

## 1 INTRODUCTION

Generative modeling is a central topic in modern deep learning research (Goodfellow et al., 2016) which finds broad applications in image processing, speech synthesis, reinforcement learning, as well as in solving inverse problems and statistical physics problems. The goal of generative modeling is to capture the full joint probability distribution of high dimensional data and generate new samples according to the learned distribution. There have been significant advances in generative modeling in recent years. Of particular interests are the variational autoencoders (VAEs) (Kingma & Welling, 2013; Rezende et al., 2014), generative adversarial networks (GANs) (Goodfellow et al., 2014), and autoregressive models (Germain et al., 2015; Kingma et al., 2016; van den Oord et al., 2016b;a; Papamakarios et al., 2017). Besides, there is another class of generative models which has so far gained less attention compared to the aforementioned models. These models invoke a sequence of diffeomorphism to connect between latent variables with a simple base distribution and data which follow a complex distribution. Examples of these flow-based generative models include the NICE and the RealNVP networks (Dinh et al., 2014; 2016), and the more recently proposed Glow model (Kingma & Dhariwal, 2018). These models enjoy favorable properties such as tractable likelihoods and efficient exact inference due to invertibility of the network.

A key concern in the design of flow-based generative models has been the tradeoff between the expressive power of the generative map and the efficiency of training and sampling. One typically needs to impose additional constraints in the network architecture (Dinh et al., 2014; Rezende & Mohamed, 2015; Kingma et al., 2016; Papamakarios et al., 2017), which unfortunately weakens the model compared to other generative models. In addition, another challenge to generative modeling is how to impose symmetry conditions such that the model generates symmetry related configurations with equal probability.

To further unleash the power of the flow-based generative model, we draw inspirations from the optimal transport theory (Villani, 2003; 2008; Peyré & Cuturi, 2018) and dynamical systems (Katok & Hasselblatt, 1995). Optimal transport theory concerns the problem of connecting two probability distributions $p(\boldsymbol{z})$ and $q(\boldsymbol{x})$ via transportation $\boldsymbol{z} \mapsto \boldsymbol{x}$ at a minimal cost. In this context, the Brenier theorem (Brenier, 1991) states that under the quadratic distance measure, the optimal generative map is the gradient of a convex function. This motivates us to parametrize the vector-valued generative map as the gradient of a scalar potential function, thereby formulating the generation process as a continuous-time gradient flow (Ambrosio et al., 2008). In this regard, a generative map is naturally viewed as a deterministic dynamical system which evolves over time. Numerical integration

of the dynamical system gives rise to the neural network representation of the generative model. To this end, E (2017) proposes a dynamical system perspective to machine learning, wherein the training procedure is viewed as a control problem, and the algorithm like back-propagation is naturally derived from the optimal control principle (LeCun et al., 1988). Moreover, Chen et al. (2018) implemented the generative map as an ODE integration and employed efficient adjoint analysis for its training.

In this paper, we devise the Monge-Ampère flow as a new generative model and apply it to two problems: density estimation of the MNIST dataset and variational calculation of the Ising model. In our approach, the probability density is modeled by a compressible fluid, which evolves under the gradient flow of a learnable potential function. The flow has tractable likelihoods and exhibits the same computational complexity for sampling and inference. Moreover, a nice feature of the Monge-Ampère flow is that one can construct symmetric generative models more easily by incorporating the symmetries into the scalar potential. The simplicity and generality of this framework allow the principled design of the generative map and gives rise to lightweight yet powerful generative models.

## 2 THEORETICAL BACKGROUND

Consider latent variables $z$ and physical variables $x$ both living in $\mathbb{R}^N$. Given a diffeomorphism between them, $x = x(z)$, the probability densities in the latent and physical spaces are related by $p(z) = q(x) \left| \det\left(\frac{\partial x}{\partial z}\right) \right|$. The Brenier theorem (Brenier, 1991) implies that instead of dealing with a multi-variable generative map, one can consider a scalar valued generating function $x = \nabla u(z)$, where the convex Brenier potential $u$ satisfies the Monge-Ampère equation (Caffarelli et al., 1998)

$$\frac{p(z)}{q(\nabla u(z))} = \det\left(\frac{\partial^2 u}{\partial z_i \partial z_j}\right). \tag{1}$$

Given the densities $p$ and $q$, solving the Monge-Ampère equation for $u$ turns out to be challenging due to the nonlinearity in the determinant. Moreover, for typical machine learning and statistical physics problems, an additional challenge is that one does not even have direct access to both probability densities $p$ and $q$. Instead, one only has independent and identically distributed samples from one of them, or one only knows one of the distributions up to a normalization constant. Therefore, solving the Brenier potential in these contexts is a control problem instead of a boundary value problem. An additional computational challenge is that even for a given Brenier potential, the right-hand side of equation 1 involves the determinant of the Hessian matrix, which scales unfavorably as $\mathcal{O}(N^3)$ with the dimensionality of the problem.

To address these problems, we consider the Monge-Ampère equation in its *linearized form*, where the transformation is infinitesimal (Villani, 2003). We write the convex Brenier potential as $u(z) = ||z||^2/2 + \epsilon\varphi(z)$, thus $x - z = \epsilon\nabla\varphi(z)$, and correspondingly $\ln q(x) - \ln p(z) = -\text{Tr}\ln\left(I + \epsilon\frac{\partial^2\varphi}{\partial z_i \partial z_j}\right) = -\epsilon\nabla^2\varphi(z) + \mathcal{O}(\epsilon^2)$. In the last equation we expand the logarithmic function and write the trace of a Hessian matrix as the Laplacian operator. Finally, taking the continuous-time limit $\epsilon \to 0$, we obtain

$$\frac{dx}{dt} = \nabla\varphi(x), \tag{2}$$

$$\frac{d\ln p(x,t)}{dt} = -\nabla^2\varphi(x), \tag{3}$$

such that $x(0) = z$, $p(x,0) = p(z)$, and $p(x,T) = q(x)$, where $t$ denotes continuous-time and $T$ is a fixed time horizon. For simplicity here, we still keep the notion of $x$, which now depends on time. The evolution of $x$ from time $t = 0$ to $T$ then defines our generative map. We notice that Chen et al. (2018) used a more general form of these two equations as a continuous-time normalizing flow (Rezende & Mohamed, 2015).

The two ordinary differential equations (ODEs) compose a dynamical system, which describes the flow of continuous random variables and the associated probability densities under iterative change-of-variable transformation. To match $p(x,T)$ and the target density $q(x)$, one can optimize a functional $I[p(x,T), q(x)]$ that measures the difference between $p(x,T)$ and $q(x)$. Thus, the training process amounts to solving an optimal control problem for the potential function:

$$\min_{\varphi} I[p(x,T), q(x)]. \tag{4}$$

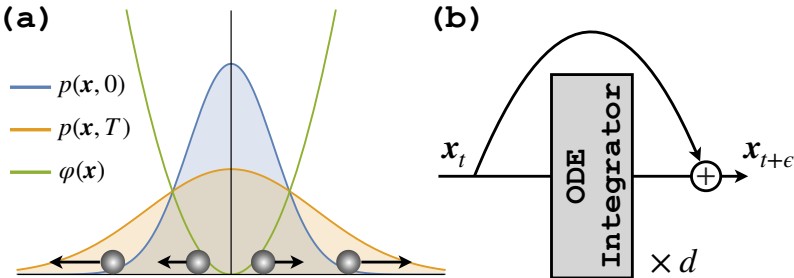

Figure 1: (a) Schematic illustration of the gradient flow of compressible fluid in one-dimension. The instantaneous velocities of fluid parcels are determined by the gradient of the potential function. The fluid density evolves to the final one after a finite flow time. (b) Numerical integration of the Monge-Ampère flow is equivalent to forward passing through a deep residual neural network. Each integration step advances these fluid parcels by one step according to the instantaneous velocity. The density distribution changes accordingly. $d$ is the total number of integration steps.

As specified later, in our examples $I[p(\boldsymbol{x}, T), q(\boldsymbol{x})]$ is defined as the Kullback-Leibler (KL) divergence or reverse KL divergence, which give rise to equations 6 and 7, respectively.

One can interpret equations 2 and 3 as fluid mechanics equations in the Lagrangian formalism. Equation 2 describes the trajectory of fluid parcels under the velocity field given by the gradient of the potential function $\varphi(\boldsymbol{x})$. While the time derivative in equation 3 is understood as the "material derivative" (Thorne & Blandford, 2017), which describes the change of the local fluid density $p(\boldsymbol{x}, t)$ experienced by someone traveling with the fluid. Figure 1(a) illustrates the probability flow in a one dimension example (Appendix A). The velocity field of the given potential pushes the fluid parcel outward, and the fluid density expands accordingly.

The fluid mechanics interpretation is even more apparent if we write out the material derivative in equation 3 as $d/dt = \partial/\partial t + d\boldsymbol{x}/dt \cdot \nabla$, and use equation 2 to obtain

$$\frac{\partial p(\boldsymbol{x}, t)}{\partial t} + \nabla \cdot [p(\boldsymbol{x}, t)\boldsymbol{v}] = 0, \tag{5}$$

which is the Liouville equation, namely, the continuity equation of a *compressible fluid* with density $p(\boldsymbol{x}, t)$ and velocity field $\boldsymbol{v} = \nabla\varphi(\boldsymbol{x})$. Obeying the continuity equation ensures that the flow conserves the total probability mass. Moreover, the velocity field is curl free $\nabla \times \boldsymbol{v} \equiv 0$ and the fluid follows a form of *gradient flow* (Ambrosio et al., 2008). The irrotational flow matches one's heuristic expectation that the flow-based generative model transports probability masses.

It should be stressed that although we use the optimal transport theory to motivate model architecture design, i.e., the gradient flow for generative modeling, we do not have to employ the optimal transport objective functions. The difference is that in generative modeling one typically fixes only one end of the probability density and aims at learning a suitable transformation to reach the other end. While for optimal transport one has both ends fixed and aims at minimizing the transportation cost. Arjovsky et al. (2017); Bousquet et al. (2017); Genevay et al. (2017) adapted the Wasserstein distances in the optimal transport theory as an objective function for generative modeling.

## 3 PRACTICAL METHODOLOGY

We parametrize the potential function $\varphi(\boldsymbol{x})$ using a feedforward neural network and integrate the ODEs (2) and (3) to transform the data and their log-likelihoods. Then, by applying the back-propagation algorithm through the ODE integration, we tune the parametrized potential function so that the probability density flows to the desired distribution [1] .

---

[1]See https://github.com/.../MongeAmpereFlow for code implementation and pre-trained models.

Figure 2: Schematic illustration of two applications (a) unsupervised density estimation and (b) variational free energy calculation for a statistical mechanics problem. In both cases, we integrate equations 2 and 3 under a parametrized potential function $\varphi(\boldsymbol{x})$, and optimize $\varphi(\boldsymbol{x})$ such that the density at the other end matches to the desired one.

In general, one can represent the potential function in various ways as long as the gradient and Laplacian are well defined. Here we adopt a densely connected neural network with only one hidden layer in our minimalist implementation. We use the softplus activation function in the hidden layers so that the potential function is differentiable to higher orders. The gradient and the Laplacian of the potential appearing in the right-hand side of equations 2 and 3 can be computed via automatic differentiation.

We integrate equations 2 and 3 using a numerical ODE integrator. At each integration step one advances the ODE system by a small amount, as shown in figure 1. Therefore, the numerical integration of ODE is equivalent to the forward evaluation of a deep residual neural network (He et al., 2016; E, 2017; Li & Shi, 2017; Chang et al., 2017; Lu et al., 2017; Rousseau & Fablet, 2018; Ruthotto & Haber, 2018; Haber & Ruthotto, 2018; Chen et al., 2018). Alternatively, the integration can also be viewed as a recurrent neural network in which one feeds the output back into the block iteratively. In both pictures, the network heavily shares parameters in the depth direction since the gradient and Laplacian information of the same potential function is used in each integration step. As a result, the network is very parameter efficient.

We employ the fourth order Runge-Kutta scheme with a fixed time step $\epsilon$, which is set such that the numerical integrator is accurate enough. Thus, the ODE integrator block shown in figure 1(b) contains four layers of neural networks. With a hundred integration steps, the whole ODE integration procedure corresponds to four hundreds layers of a neural network. At the end of the integration, we obtain samples $\boldsymbol{x}$ and their likelihoods $\ln p(\boldsymbol{x}, T)$, which depends on the parametrized potential function $\varphi(\boldsymbol{x})$. Differentiable optimization of such deep neural network is feasible since the integrator only applies small changes to the input signal of each layer.

The deep generative model resulting from the discretization of an ODE system shows some nice features compared to the conventional neural networks. At training time, there is then a tradeoff between the total number of integration steps and the expressibility of the potential function. Longer integration time corresponds to a deeper network, and hence a simpler transformation at each step. While at testing time, one can construct a variable depth neural network for the potential function. For example, one can use a larger time step and a smaller number of integration steps for efficient inference. Moreover, by employing a reversible ODE integrator, one can also integrate the equations 2 and 3 backward in time with the same computational complexity, and return to the starting point deterministically.

## 4 APPLICATIONS

We apply the Monge-Ampère flow to unsupervised density estimation of an empirical dataset and variational calculation of a statistical mechanics problem. Figure 2 illustrates the schemes of the two tasks. To draw samples from the model and to evaluate their likelihoods, we simulate the fluid dynamics by integrating the ODEs (2) and (3). We use the KL divergence as the measure of dissimilarity between the model probability density and the desired ones. Moreover, we choose the base distribution at time $t = 0$ to be a simple Gaussian $p(\boldsymbol{x}, 0) = \mathcal{N}(\boldsymbol{x})$. See Appendix C for a summary of the hyperparameters used in the experiments.

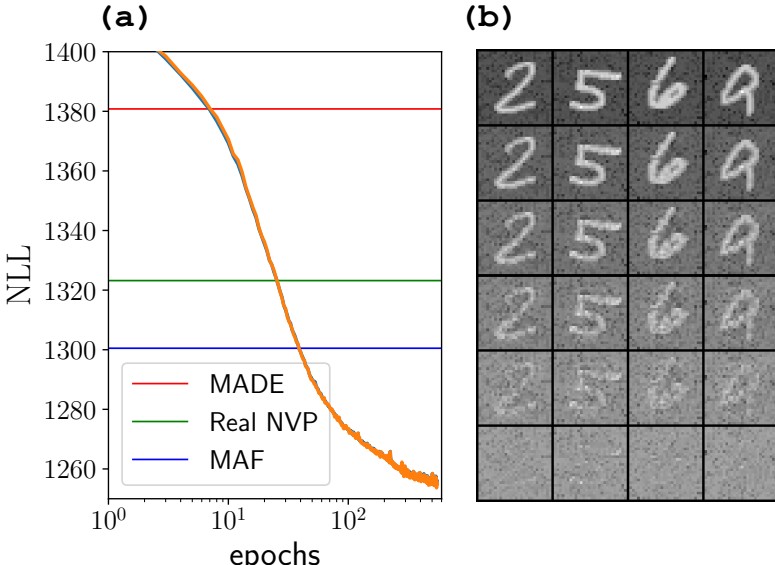

Figure 3: (a) The NLL of the training (blue) and the test (orange) MNIST dataset. The horizontal lines indicate results obtained with previous flow-based models reported in (Papamakarios et al., 2017). (b) From top to bottom, Monge-Ampère flow of test MNIST images to the base Gaussian distribution.

## 4.1 DENSITY ESTIMATION ON THE MNIST DATASET

First we perform the maximum likelihood estimation, which reduces the dissimilarity between the empirical density distribution $\pi(\boldsymbol{x})$ of a given dataset and the model density $p(\boldsymbol{x}, T)$ measured by the KL-divergence $D_{\mathrm{KL}}(\pi(\boldsymbol{x})\|p(\boldsymbol{x}, T))$. It is equivalent to minimize the negative log-likelihood (NLL):

$$\mathrm{NLL} = -\mathbb{E}_{\boldsymbol{x}\sim\pi(\boldsymbol{x})}[\ln p(\boldsymbol{x}, T)]. \tag{6}$$

To compute the model likelihood we start from MNIST samples and integrate backwards from time $T$ to 0. By accumulating the change in the log-likelihood $\int_T^0 d\ln p(\boldsymbol{x}(t), t)$ we obtain an estimate of the objective function in equation 6.

To model the MNIST data we need to transform the images into continuous variables. Following (Papamakarios et al., 2017), we first apply the jittering and dequantizing procedure to map the original grayscale MNIST data to a continuous space. Next, we apply the logit transformation to map the normalized input to the logit space $\boldsymbol{x} \mapsto \mathrm{logit}(\lambda + (1 - 2\lambda)\boldsymbol{x})$, with $\lambda = 10^{-6}$. Figure 3(a) shows the training and test NLL compared with the best performing MADE (Germain et al., 2015), Real-NVP (Dinh et al., 2016), and MAF (Papamakarios et al., 2017) models, all reported in Papamakarios et al. (2017). Note that we have selected the results with standard Gaussian base distribution for a fair comparison. The test NLL of the present approach is lower than previously reported

Table 1: Average test NLL for density estimation on MNIST. Lower is better. The benchmark results are quoted from the Table 2 of (Papamakarios et al., 2017).

| Model | Test NLL |
|---|---|
| MADE | $1380.8 \pm 4.8$ |
| Real NVP | $1323.2 \pm 6.6$ |
| MAF | $1300.5 \pm 1.7$ |
| **Present** | $\mathbf{1255.5 \pm 2.0}$ |

values, see Table 1 for a summary. The Monge-Ampère flow is quite parameter-efficient as it only uses about one-tenth of learnable parameters of the MAF model (Papamakarios et al., 2017).

The way we carry out density estimation in figure 2(a) is equivalent to transforming the data distribution $\pi(\boldsymbol{x})$ to the base Gaussian distribution (Papamakarios et al., 2017). Figure 3(b) shows the flow from given MNIST images to the Gaussian distribution. One observes that the flow removes features in the MNIST images in a continuous manner. The procedure is a continuous-time real-

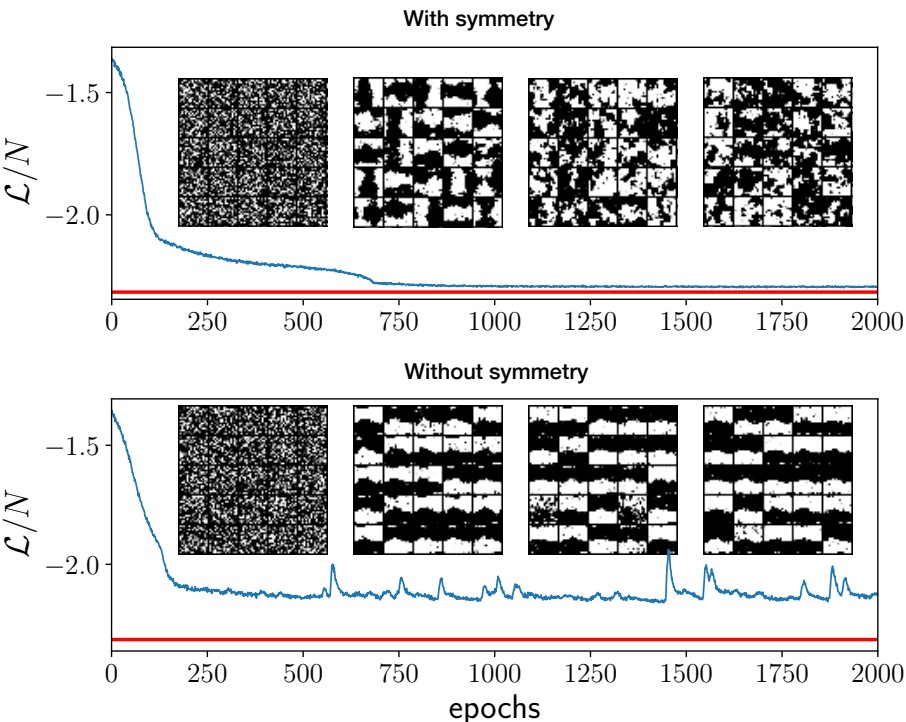

Figure 4: The training loss equation 7 vs the training epoch with (upper panel) and without (lower panel) translational and rotational symmetry imposed in the potential. The loss function is the variational free energy of the Ising model, which is lower bounded by the exact solution indicated by the horizontal red line (Onsager, 1944; Li & Wang, 2018). Inset shows representative Ising configurations generated at epochs $0, 500, 1000, 1500$, respectively. Each inset contains a $5 \times 5$ tile of a $16^2$ Ising model. In the lower panel, the generated samples exhibit two domains because we still impose the $\mathbb{Z}_2$ inversion symmetry in the network. Without imposing the inversion symmetry the model will generate almost all black/white images due to the ferromagnetic correlations.

ization of the Gaussianization technique (Chen & Gopinath, 2001). Conversely, by integrating the equations forward in time one can use the flow to map Gaussian noises to meaningful images.

## 4.2 VARIATIONAL LEARNING FOR STATISTICAL MECHANICS PROBLEM

For variational calculation we minimize the reverse KL divergence between the model and a given Boltzmann distribution of a physical problem $D_{\mathrm{KL}}\left(p(\boldsymbol{x}, T) \| \frac{e^{-E(\boldsymbol{x})}}{Z}\right)$, where $Z = \int d\boldsymbol{x} e^{-E(\boldsymbol{x})}$ is an unknown partition function due to its intractability. In practical this amounts to minimizing the following expectation

$$\mathcal{L} = \mathbb{E}_{\boldsymbol{x} \sim p(\boldsymbol{x}, T)} \left[\ln p(\boldsymbol{x}, T) + E(\boldsymbol{x})\right]. \tag{7}$$

To draw samples from $p(\boldsymbol{x}, T)$ we start from the base distribution $\boldsymbol{x} \sim p(\boldsymbol{x}, 0) = \mathcal{N}(\boldsymbol{x})$ and evolve the samples according to equation 2. We keep track of the likelihoods of these samples by integrating equation 3 together. The loss function equation 7 provides a variational upper bound to the physical free energy $-\ln Z$, since $\mathcal{L} + \ln Z = D_{\mathrm{KL}}\left(p(\boldsymbol{x}, T) \| \frac{e^{-E(\boldsymbol{x})}}{Z}\right) \geq 0$. Differentiable optimization of the objective function employs the reparametrization pathwise derivative for the gradient estimator, which exhibits low variance and scales to large systems (Kingma & Welling, 2013).

We apply this variational approach to the Ising model, a prototypical problem in statistical physics. The Ising model exhibits a phase transition from ferromagnetic to paramagnetic state at the critical temperature (Onsager, 1944). Variational study of the Ising model at the critical point poses a stringent test on the probabilistic model. To predict the thermodynamics accurately, the flow has to capture long-range correlations and rich physics due to the critical fluctuation. In the continuous

representation (Fisher, 1983; Zhang et al., 2012; Li & Wang, 2018), the energy function of the Ising model reads (Appendix B)

$$E(\boldsymbol{x}) = \frac{1}{2}\boldsymbol{x}^T K^{-1} \boldsymbol{x} - \sum_i \ln \cosh(x_i). \tag{8}$$

We set the coupling $K_{ij} = (1 + \sqrt{2})/2$ to be at the critical temperature of square lattice Ising model (Onsager, 1944), where $i, j$ are nearest neighbors on the square periodic lattice. The expectation inside the bracket of equation 7 is the energy difference between the model and the target problem. Since one has the force information for both target problem and the network as well, one can introduce an additional term in the loss function for the force difference (Czarnecki et al., 2017; Zhang et al., 2018a).

The Ising model on a periodic square lattice has a number of symmetries, e.g., the spin inversion symmetry, the spatial translational symmetry, and the $D_4$ rotational symmetry. Mathematically, the symmetries manifest themselves as an invariance of the energy function $E(\boldsymbol{x}) = E(\mathbf{g}\boldsymbol{x})$, where g is the symmetry operator that inverts or permutes the physical variables. For physics applications, it is essential to have a generative model that respects these physical symmetries, such that they will generate symmetry-related configurations with equal probability. Although simple symmetries such as the $Z_2$ inversion can be implemented by careful design of the network or simply averaging over the elements in the symmetry group (Li & Wang, 2018), respecting even more general symmetries in a generative model turns out to be a challenging problem. The Monge-Ampère flow naturally solves this problem by imposing the symmetry condition on the scalar potential function since the initial Gaussian distribution respects the maximum symmetry.

There have been a number of studies about how to ensure suitable symmetries in the feedforward networks in the context of discriminative modeling (Zaheer et al., 2017; Han et al., 2018b; Cohen et al., 2018; Zhang et al., 2018b). Here we employ a simpler approach to encode symmetry in the generating process. We express the potential as an average over all elements of the symmetry group $G$ under consideration $\varphi(\boldsymbol{x}) = \frac{1}{|G|}\sum_{\mathbf{g}\in G}\tilde{\varphi}(\mathbf{g}\boldsymbol{x})$, where the $\tilde{\varphi}$ in each term shares the same network parameters. At each step of the numerical integration, we sample only one term in the symmetric average to evaluate the gradient and the Laplacian of the potential. Thus, on average one restores the required symmetry condition in the generated samples. This is feasible both for training and data generation since equations 2 and 3 are both linear with respect to the potential function.

The upper panel of Figure 4 shows that the variational loss equation 7 decreases towards the exact solution of the free energy (Appendix B). The achieved $1\%$ relative accuracy in the variational free-energy is comparable to the previously reported value ($0.5\%$) using a network which exploits the two-dimensional nature of the problem (Li & Wang, 2018). The inset shows the directly generated Ising configurations from the network at different training stage. At late stages, the network learned to produce domains of various shapes. One also notices that the generated samples exhibit a large variety of configurations, which respects the physical symmetries. Moreover, the network can estimate log-likelihoods for any given sample which is valuable for the quantitative study of the physical problem. The invertible flow can also perform inverse mapping of the energy function from the physical space to the latent space. Therefore, it can be used to accelerate Monte Carlo simulations, by either recommending the updates as a generative model (Huang & Wang, 2017; Liu et al., 2017) or performing hybrid Monte Carlo in the latent space (Li & Wang, 2018).

Figure 4 also compares the performances of the generative map with and without translational and rotational symmetries imposed. It is noticeable that using the same parameters in the network and the same training strategy, the variational free energy of the one without symmetry constraint is significantly higher than the one with symmetry constraint. Moreover, by inspecting the generated samples, one sees that the model distribution collapses to the ones with two horizontal domains. These symmetry breaking configuration correspond to a metastable local minimum of the free energy landscape. This comparison shows that imposing the necessary symmetry conditions is crucial for variational studies of physical problems using generative models

## 5 RELATED WORK

**Normalizing flows:** The present approach is closely related to the normalizing flows, where one composes bijective and differentiable transformations for generative modeling (Rezende & Mo-

hamed, 2015). To scale to large dataset it is crucial to make the Jacobian determinant of the transformation efficiently computable. Dinh et al. (2014; 2016); Li & Wang (2018); Kingma & Dhariwal (2018) strive the balance between expressibility and efficiency of the bijective networks by imposing block structures in the Jacobian matrix. Our approach reduces the cubical scaling Hessian determinant to the linear scaling Laplacian calculation by taking the continuous-time limit. The continuous-time gradient flow is more flexible in the network design since bijectivity and the efficiently tractable Jacobian determinant is naturally ensured.

**(Inverse) autoregressive flows:** The autoregressive property could be interpreted as imposing a triangular structure in the Jacobian matrix of the flow to reduce its computational cost. There is a trade-off between the forward and inverse transformation. Therefore, typically the autoregressive flows are used for density estimation (Papamakarios et al., 2017), while the inverse autoregressive flows are suitable for posterior in variational inference (Kingma et al., 2016). Moreover, these flows are only implicitly reversible, in the sense that the inverse calculation requires solving a nonlinear equation. However, the continuous-time gradient flow has the same time complexity for both the forward and inverse transformation.

**Continuous normalizing flows:** Chen et al. (2018) derived the continuous-time limit of the normalizing flows Rezende & Mohamed (2015), which is a general form of equations 2 and 3 without the irrational condition on the velocity field. Besides conceptual connection to the Brenier theorem in optimal transport theory, the motivation of using the gradient flow instead of more general transformations in Chen et al. (2018) is that it is convenient to encode symmetries in the scalar potential function. Furthermore, Grathwohl et al. (2018) made use of the Hutchinson's trace estimator (Hutchinson, 1990) to simplify the computation of the trace of Jacobian, which we could also employ for the Laplacian in equation (3). Salman et al. (2018) investigated regularization of such continuous-time flow.

**(Continuous-time) diffusive flows**: Tabak & Vanden-Eijnden (2010); Sohl-Dickstein et al. (2015); Chen et al. (2017); Mesa et al. (2018); Frogner & Poggio (2018) consider generative models based on diffusion processes. Our work is different since we consider deterministic and reversible advection flow of the fluid. There is, in general, no stochastic force in our simulation (except the random sampling of symmetric functions done in section 4.2). And, we always integrate the flow equation for a finite time interval instead of trying to reach a steady probability density in the asymptotic long time limit.

**Dynamical system and control based methods**: In a more general perspective, our approach amounts to defining a dynamical system with a target terminal condition, which is naturally viewed as a control problem (E, 2017). Along with this direction, Han & E (2016); Han et al. (2018a); Li et al. (2017) offer some insights into interpreting and solving this control problem. Moreover, the Neural ODE (Chen et al., 2018) implements an efficient back-propagation scheme through black-box ODE integrators based on adjoint computation, which we did not utilize at the moment.

## 6 DISCUSSIONS

Gradient flow of compressible fluids in a learnable potential provides a natural way to set up deep generative models. The Monge-Ampère flow combines ideas and techniques in optimal transport, fluid dynamics, and differential dynamical systems for generative modeling.

We have adopted a minimalist implementation of the Monge-Ampère flow with a scalar potential parameterized by a single hidden layer densely connected neural network. There are a number of immediate improvements to further boost its performance. First, one could extend the neural network architecture of the potential function in accordance with the target problem. For example, a convolutional neural network for data with spatial or temporal correlations. Second, one can explore better integration schemes which exactly respect the time-reversal symmetry to ensure reversible sampling and inference. Lastly, by employing the backpropagation scheme of Chen et al. (2018) through the ODE integrator one can reduce the memory consumption and achieve guaranteed convergence in the integration.

Furthermore, one can employ the Wasserstein distances (Arjovsky et al., 2017) instead of the KL-divergence to train the Monge-Ampère flow. With an alternative objective function, one may obtain an even more practically useful generative model with tractable likelihood. One may also consider

using batch normalization layers during the integration of the flow (Dinh et al., 2016; Papamakarios et al., 2017). However, since the batch normalization breaks the physical interpretation of the continuous gradient flow of a fluid, one still needs to investigate whether it plays either a theoretical or a practical role in the continuous-time flow.

Moreover, one can use a time-dependent potential $\varphi(\boldsymbol{x}, t)$ to induce an even richer gradient flow of the probability densities. Benamou & Brenier (2000) has shown that the optimal transport flow (in the sense of minimizing the spatial-time integrated kinetic energy, which upper bounds the squared Wasserstein-2 distance) follows a *pressureless* flow in a time-dependent potential. The fluid moves with a constant velocity that linearly interpolates between the initial and the final density distributions. Practically, a time-dependent potential corresponds to a deep generative model without sharing parameters in the depth direction as shown in figure 1(b). Since handling a large number of independent layers for each integration step may be computationally inefficient, one may simply accept one additional time variable in the potential function, or parametrize $\varphi(\boldsymbol{x}, t)$ as the solution of another differential equation, or partially tie the network parameters using a hyper-network (Ha et al., 2016).

Besides applications presented here, the Monge-Ampère flow has wider applications in machine learning and physics problems since it inherits all the advantages of the other flow-based generative models (Dinh et al., 2014; Rezende & Mohamed, 2015; Kingma et al., 2016; Dinh et al., 2016; Papamakarios et al., 2017). A particular advantage of generative modeling using the Monge-Ampère flow is that it is relatively easy to impose symmetry into the scalar potential. It is thus worth exploiting even larger symmetry groups, such as the permutation for modeling exchangeable probabilities (Korshunova et al., 2018). Larger scale practical application in statistical and quantum physics is also feasible with the Monge-Ampère flow. For example, one can study the physical properties of realistic molecular systems using Monge-Ampère flow for variational free energy calculation. Lastly, since the mutual information between variables is greatly reduced in the latent space, one can also use the Monge-Ampère flow in conjunction with the latent space hybrid Monte Carlo for efficient sampling (Li & Wang, 2018).

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

## A  SOLUTION OF A 1-D GAUSSIAN DISTRIBUTION

To gain intuition about the probability density flow under the Monge-Ampère equation, we work out a close form solution of equations 2 and 3 in a one dimensional toy problem.

Since a quadratic potential merely induces scale transformation to the data, a Gaussian distribution will remain to be a Gaussian with a different variance. Thus, we consider $\varphi(x) = \lambda x^2/2$, and $p(x,0) = \mathcal{N}(x)$, and parametrize the time dependent density as $p(x,t) = \frac{\alpha(t)}{\sqrt{2\pi}} \exp\left(-\frac{\alpha(t)^2 x^2}{2}\right)$, and $\alpha(0) = 1$. Substitute this ansatz of the time-dependent probability density into equation 3, we have

$$
\begin{aligned}
-\lambda = \frac{d\ln p(x,t)}{dt} &= \frac{\partial \ln p(x,t)}{dt} + \frac{dx}{dt}\frac{\partial \ln p(x,t)}{\partial x} \\
&= \frac{d\ln\alpha}{dt} - \left(\frac{d\ln\alpha}{dt} + \lambda\right)\alpha^2 x^2,
\end{aligned}
\tag{9}
$$

where we used equation 2, i.e. $\frac{dx}{dt} = \lambda x$ in the second line. Thus, $\frac{d\ln\alpha}{dt} = -\lambda$, and $\alpha(t) = \exp(-\lambda t)$. Therefore, under a quadratic potential the fluid parcel moves with acceleration. The ones that are far away from the origin move faster. And the width of the Gaussian distribution changes exponentially with time.

## B  CONTINUOUS FORMULATION OF THE ISING MODEL

The Ising model is a fundamental model in statistical mechanics which allows exact solution in two dimension (Onsager, 1944). The Ising model partition function reads

$$
Z_{\text{Ising}} = \sum_{s \in \{\pm 1\}^{\otimes N}} \exp\left(\frac{1}{2}s^T K s\right).
\tag{10}
$$

To make it amenable to the flow approach we reformulate the Ising problem to an equivalent representation in terms of continuous variables following (Fisher, 1983; Zhang et al., 2012; Li & Wang, 2018). First, we offset the coupling to $K + \alpha I$ such that all of its eigenvalues are positive, e.g. the minimal eigenvalue of $K + \alpha I$ is 0.1. This step does not affect the physical property of the Ising model except inducing a constant offset in its energies. Next, using the Gaussian integration trick (Hubbard-Stratonovich transformation) we can decouple the Ising interaction with continuous auxiliary variables $Z_{\text{Ising}} \propto \sum_s \int dx \exp\left(-\frac{1}{2}x^T(K+\alpha I)^{-1}x + s^T x\right)$. Finally, tracing out Ising variables we arrived at the energy function equation 8 for the continuous variables in the main texts. After solving the continuous version of the Ising problem, one can obtain the original Ising configuration via direct sampling from $p(s|x) = \prod_i \sigma(2x_i)$.

The free energy of the associated continuous representation of the Ising model reads $-\ln Z = -\ln Z_{\text{Ising}} - \frac{1}{2}\ln\det(K+\alpha I) + \frac{N}{2}[\ln(2/\pi) - \alpha]$, where the analytical expression of $Z_{\text{Ising}}$ on finite periodic lattice can be obtained from Eq. (39) of Kaufman (1949).

## C  HYPERPARAMETERS

We list the hyperparameters of the network and training in Table 2. $\epsilon$ in the integration step in the fourth order Runge-Kutta integration. $d$ is the total number of integration steps. Thus $T = \epsilon d$ is the total integration time. $h$ is the number of hidden neurons in the potential function. $B$ is the mini-batch size for training.

Table 2: Hyperparameters of experiments reported in figure 3 and figure 4.

| Problem | $\epsilon$ | $d$ | $h$ | $B$ |
|---|---|---|---|---|
| MNIST | 0.1 | 100 | 1024 | 100 |
| Ising | 0.1 | 50 | 512 | 64 |

