# OpenReview forum: "Monge-Amp\`ere Flow for Generative Modeling"
_ICLR.cc/2019/Conference_

### Official Review · AnonReviewer3 · 2018-11-01
**Proposes a novel parameter-efficient generative modeling approach based on the Monge-Ampere equation. Applications section is not convincing enough.**

**Rating:** 6
**Confidence:** 3

**Review:**

This paper proposes a novel parameter-efficient generative modeling approach that is based on the Monge-Ampere equation. In the proposal, a feed-forward neural network is trained as an ODE integrator which solves (2) and (3) for a fixed time interval $[0,T]$, so that the distribution $p(x,t)$ at time 0 is a simple base distribution such as a Gaussian, and that at time $T$ mimics the target distribution.

[pros]
- The proposal provides a parameter-efficient approach to generative modeling, via parameter sharing in the depth direction.
- I think that the idea itself is quite interesting and that it is worth pursuing this direction further.

[cons]
- The Applications section is not convincing enough to demonstrate usefulness of the proposal as an approach to generative modeling.
- How the gradient-based learning in the proposal behaves is not discussed in this paper.

[quality]
How the gradient-based learning in the proposal behaves is not discussed. I understand that the non-convex nature of the loss function poses problems already in the conventional back-propagation learning of a multilayer neural network. On the other hand, in the proposal, the loss function (e.g., (4)) is further indirectly parameterized via $\varphi$. It would be nice if the parameterization of the loss in terms of $\varphi$ is regular in some sence.

[clarity]
Description of this paper is basically clear. In the author-date citation style employed in this paper, both the author names and publication year are enclosed in parentheses, with exception being the author names incorporated in the text. This paper does not follow the above standard convention for citation and thus poses strong resistance to the reader. For example, in the first line of the Introduction section, "Goodfellow et al. (2016)" should read "(Goodfellow et al., 2016)".

[originality]
The idea of considering the Monge-Ampere equation in its linearized form to formulate generative modeling seems original.

[significance]
In the experiment described in Section 4.1, it is not clear at all from the description here whether the learned system is capable of successfully generating MNIST-like fake images, which would question the significance of the proposal as a framework for generative modeling. It is well known that the KL divergence $D(P\|Q)$ tends to put more penalty when $P$ is large and $Q$ is small than the opposite. One can then expect in this experiment that it tolerates the model, appearing as $Q$ in $D(P\|Q)$, to put weights on regions where the data are scarce, which might result in generation of low-quality fake images. It would be nice if the authors provide figures showing samples generated via mapping of Gaussian samples with the learned system.
Also, in the experiment described in Section 4.2, I do not see its significance. It is nice to observe in Figure 4 that the loss function approaches the true free energy as well as that the snapshots generated by the model seem more or less realistic. My main concern however is regarding what the potential utilities of the proposal are in elucidating statistical-physical properties of a system. For example, it would be nice if the proposal could estimate the phase-transition point more easily and/or more accurately compared with alternative conventional approaches, but there is no such comparison presented in this paper, making the significance of this paper obscure.

Minor points:

The reference entitled "A proposal on machine learning via dynamical systems" would be better cited not as "E (2017)" but rather as "Weinan (2017)".

Page 6, line 10: the likelihoods of these sample(s)

----Updated after author feedback----
Upon reading the author feedback, I have downgraded my rating from 7 to 6, because the author feedback is not satisfactory to me in some respects. In my initial review, my comment on the experiment on MNIST is not on correlation between the maximum likelihood estimation and visual quality of generated images, on which the author feedback was based, but regarding the well-known property of the KL divergence due to its asymmetry between the two arguments. Also, regarding the experiment on the Ising model, the proposal in this paper provides an approximate sampler, whereas for example the MCMC provides an exact sampler with exponential slowing down in mixing under multimodal distributions. In statistical physics, one is interested in studying physical properties of the system, such as phase transition, with samples obtained from a sampler. In this regard, important questions are how good the samples are and how efficiently they are generated. As for the quality, it would have been nice if results of evaluated free energy as a function of inverse temperature (that is K_ij in the case here) were provided. The author feedback was, on the other hand, mainly explanation of general variational approach, of which I am aware.
I still think that this paper contains interesting contributions, and accordingly have put my rating above the threshold.

---

> ### Author Response · Authors · 2018-11-21
> **Good Questions! More details regarding loss function of density estimation and advantage for statistical physics problem are given**
>
> > How the gradient-based learning in the proposal behaves is not discussed ... It would be nice if the parameterization of the loss in terms of $\varphi$ is regular in some sence.
>
> Nice observation! At this moment we do not have a definite conclusion to this problem, but there are several empirical observations. First, let us consider the dynamical system structure of the proposed construction. Such construction resembles the ResNet, whose learning ability (representability), regularity, and optimization have been a hot topic. See, e.g., https://arxiv.org/pdf/1611.04231.pdf, and https://arxiv.org/abs/1603.05027. The skip connection, which is essentially the time discretization in the proposed scheme, seems to empirically improve the loss landscape. This could also be the reason why we are able to do an optimization for the network of 400 layer depth. In addition, the gradient structure adds two more ingredients. First, it guarantees some optimality of the function to be parameterized, which is derived from the optimal transport theory. Second, with symmetry constraint imposed, the regularity is further improved.
>
>
> > ... citation format ...
>
> Thanks! Revised!
>
> > ... It would be nice if the authors provide figures showing samples generated via mapping of Gaussian samples with the learned system
>
> It is a standard practice to use forward KL divergence to train the generative models with explicit and tractable likelihoods. Our training protocol is the same as other flow-based generative models. Indeed, we agree that the visual quality of samples may not correlate well with the maximum likelihood estimation objective function, for the reasons mentioned by the Referee.  By adopting training schemes such as GAN may help to address the issue raised by the Referee. We would like to perform these studies in the future.
>
> Nevertheless, here are some samples generated from the model after 500 epochs of training: https://mongeanpereflow.tumblr.com/post/180329851804/images-generated-by-monge-ampere-flow.
> We notice that we are using a simple network structure without convolutional layers.
>
> The reason that we are not confident enough to show them in the manuscript is that the generation involves integration the ODEs from $t=0$ to $T$, while for the training of density estimation we integrate the ODEs from $t=T$ to $0$.  Thus, the above result could be corrupted due to not ensuring the time-reversibility in the Runge-Kutta integrator. Further investigation of using strictly reversible integrator (such as Leapfrog and Verlet) would resolve this part of the issue.
>
> > ... My main concern however is regarding what the potential utilities of the proposal are in elucidating statistical-physical properties of a system. For example, it would be nice if the proposal could estimate the phase-transition point more easily and/or more accurately compared with alternative conventional approaches, but there is no such comparison presented in this paper, making the significance of this paper obscure.
>
> Variational free-energy calculation is a standard way of solving a statistical physics problem. For example, via computing correlations on the directly generated samples one can directly obtain physical observables such as magnetization, specific heat, and susceptibility etc. These observables signify the phase transition and the physical properties of the system.
>
> When compared to alternative conventional approaches, e.g. the  Markov chain Monte Carlo method (MCMC) approach, the variational approach has the advantage of directly accessing the free-energy and entropy of the system. Moreover, MCMC suffers from autocorrelation for high dimensional multimode distributions and the critical slowing down, the variational approach is more promising to scale up to large problems which may contain millions of variables with complex landscape.
>
> Historically, the variational approach was first developed in statistical physics, then adopted by machine learning and graphical models community as variational inference. Our contribution being taking the latest  flow-based generative model as an effective variational ansatz for classic statistical mechanics problems. And we demonstrated that it is possible, and in fact crucial to impose the physical symmetry in such variational calculation.
>
> > The reference entitled "A proposal on machine learning via dynamical systems" would be better cited not as "E (2017)" but rather as "Weinan (2017)".
> >
> > Page 6, line 10: the likelihoods of these sample(s)
>
> Thanks for the careful reading! We fixed the typo on Page 6.
>
> For the reference, "E" is indeed the last name of the author.  "Weinan" is his first name. So we keep the citation as "(E, 2017)".  Thank you!

---

> ### Author Response · Authors · 2018-11-27
> **Sorry for misunderstanding and will address more**
>
> > ... my comment on the experiment on MNIST is not on correlation between the maximum likelihood estimation and visual quality of generated images, ..., but regarding the well-known property of the KL divergence due to its asymmetry between the two arguments.
>
> Sorry for misunderstanding your previous comments. We fully agree with your comments.
> The forward and reverse KL divergence has a different tradeoff. For the density estimation problem, we minimize the forward KL (MLE), while for the variational calculation we minimize the reverse KL divergence. The choice for the later one is that we aim to solve statistical physics problems without the need to resort other meaning of sampling.
> As the Referee pointed out, minimizing the reverse KL divergence has a caveat that the variational distribution is typically narrower than the target distribution. This corresponds to the local minimum of the reverse KL divergence. The symmetry breaking solution showed in Fig. 4 is in one such example. We partially addressed this issue by imposing physical symmetries. Hope this again highlights the importance of symmetry considerations.
> In any case, the general problem of the KL divergence, as the Referee has pointed out, is still there. We shall investigate some other optimization strategies.
>
> > Also, regarding the experiment on the Ising model, the proposal in this paper provides an approximate sampler, whereas for example the MCMC provides an exact sampler ...
>
> Sorry for those overly general remarks in last correspondence. We presumed that those were not clear.
> Right. the flow-based generative model is an “approximate sampler” which supports direct parallel sampling, while MCMC is an “exact sampler” in the asymptomatic long time limit. Our motivation is that for large-scale problems with complex landscapes the variational approach outweighs the MCMC approach for practical matter (known as the variational inference of probabilistic graphical models). Thanks for your comments.
>
> We will add the results of scanning temperature in the final manuscript.
>
> We'd also like to point out and investigate more that
> 1. The network outputs are good MCMC update proposals for rejection sampling,
> 2. Exploiting the reversibility of the network, one can derive the energy function in the latent space and perform accelerated latent space MCMC.
> Both approaches give unbiased samples, where the variational ansatz provides a good starting point. We noted these in Sec. 4.2.

---

### Official Review · AnonReviewer2 · 2018-11-02
**Interesting but needs more work: clarifications to existing work and ablations would make this paper more appealing**

**Rating:** 6
**Confidence:** 4

**Review:**

This paper proposes a continuous-time gradient flow as an alternative normalizing flow. The formulation is motivated from optimal transport and the Monge-Ampere equation. Symmetry constraints motivated by the use of a potential function can be enforced during training.

I'm surprised [1] was mentioned only for their backpropagation scheme. Much of this paper is similar to theirs, such as Eqs (2) and (3) being the "instantaneous change of variables" in [1], the mention of efficient reversibility, experimenting with forward and reverse KL objectives, and parameter efficiency.

Given the different angle of approach in this work, I'm willing to believe some of this is independently done. This work contains interesting derivations and a different parameterization, with enough contributions to potentially be interesting in its own right. However, I firmly believe in proper attribution and believe [1] should at least be mentioned in the introduction and/or theoretical background.

Pros:
 - The potential function and the resulting gradient flow parameterization is interesting.
 - Parameterizing a potential function motivates some symmetry constraints.
 - Interesting application of normalizing flows to the Ising model.

Cons:
 - Paper presentation needs some work on clarity.
 - Relation to existing work needs to be clarified.
 - Experiments lack ablations and proper comparisons. e.g. the effect of using symmetry constraints, the effect of using a gradient flow parameterization.
 - If I understood correctly, the symmetry "constraints" are really data augmentation during the training phase, rather than hard constraints on the model.

Main questions:
- It seems the potential function plays a similar role to the negative log-likelihood in [2].
- Does having symmetry constraints lead to a better model when the constraints are justified? ie. can you provide comparisons for the Ising model experiment in 4.2?
- What are the set of constraints you can specify using a potential function? Permutation of the variables is very briefly mentioned in the experiment section, but this could be clarified much earlier.
- I may have missed this, but what exactly are the symmetry conditions that were used in the experiments?
- It seems that the proposed permutation constraints could be part of the training algorithm rather than the model. How different would it be if you permute the data samples and use an existing normalizing flow algorithm? ie. can you provide comparisons where randomly permuted data samples are also used during training with existing algorithms?
- Since you used a fixed-step size solver, do you have some guarantees, theoretical or empirically, that the numerical integration has low error? e.g. what is the reconstruction error from doing a forward and reverse pass, and what would the error be if compared to a much smaller step size?

Minor:
- The potential function is parameterized directly but is not integrated to infinity. Since the resulting gradient flow is time-invariant, how this would affect the expressivity of the flow? Could a time-variant potential function be used?
- Eq (5) is also the Liouville equation, which I think should be mentioned.
- MNIST digits have completely black backgrounds, so I don't understand why Figure 3 samples have grey backgrounds. Could this have something to do with numerical issues in reversing the numerical integration?
- It's awkward that Figure 3 contains the loss over training for Monge-Ampere flows but only the final loss for the rest. Table 1 sufficiently summarizes this figure, so unless you can show the loss over training for all methods I think this figure is redundant.
- Equations are referenced both with and without parenthesis. It'd be best if this is consistent across text.
- There are quite a few grammar mistakes, especially around important digression. (e.g. top of page 3 "experienced by someone travels with the fluid" -> "experienced by someone traveling with the fluid".)
- Please use citep and citet properly. Many references should be done using citep (with brackets around the author-year), when the author is a not a part of the sentence.

[1] Chen, Tian Qi, et al. "Neural Ordinary Differential Equations."
[2] Tabak, Esteban G., and Eric Vanden-Eijnden. "Density estimation by dual ascent of the log-likelihood."

---

> ### Author Response · Authors · 2018-11-21
> **More careful clarifications to existing work and additional ablation studies on imposing symmetry**
>
> >  ... However, I firmly believe in proper attribution and believe [1] should ...
>
> In the revised manuscript we cited [1] more prominently in sections 1 and 2. We also added a paragraph in Related works on Neural ODE and https://arxiv.org/abs/1810.01367.
>
> > ... the potential function plays a similar role to the negative log-likelihood in [2].
>
> The continuous-time formalism in [2] is related but we would not say its negative log-likelihood plays a similar role. The major difference is that Tabak et al directly derived a variational equation for the generative map (equation 2.2 of [2]) and turned the problem into solving a PDE. The nonlinear diffusion equation reaches the target distribution as the stationary solution. We constructed a deterministic and reversible flow of the fluid. So we always integrate the flow equation for a finite time interval instead of trying to reach a steady solution in the long time limit.
>
> More concisely, the log-likelihood $\mu(x)$ in [2] corresponds to the fixed base density. Equation 2.5 in [2] computes the velocity field by comparing the current and target densities, which is different from what we did: learning the gradient field via back propagation.
>
> > ... having symmetry constraints lead to a better model? ...
>
> Yes. Without imposing the symmetries the training falls into a local minimum of the free energy. Please see reply to Reviewer 1's question and the new Fig. 4 and corresponding discussions in the new manuscript.
>
> > ...What are the set of constraints you can specify using a potential function? ...
>
> For the Ising model, we impose the $\mathbb{Z}_2$ symmetry by using $\ln\cosh$ activation function. We impose the spatial translational/rotational symmetry by adding together all group elements for the potential function.
>
> >  .... what exactly are the symmetry conditions ...?
>
> We did not impose symmetry for the density estimation problem.
> For the Ising model, we employed the $\mathbb{Z}_2$ inversion symmetry, spatial translational symmetry, and $C_4$ rotational symmetry of the square lattice.
>
> > ... How different would it be if you permute the data samples and use an existing normalizing flow algorithm? ...
>
> Good question!
>
> Let us first clarify that our symmetry considerations are intended for the **variational calculation** other than the **density estimation**. We agree that randomly permuted data in the training procedure works fine for density estimation. However, a similar idea does not apply well to variational calculations. The closest attempt is the symmetric variational calculation in https://arxiv.org/abs/1802.02840 (equation S2), which amounts to setting up a mixture model $\ln p(x) = \ln \left[\frac{1}{|G|}\sum_{\mathtt{g}\in G} \tilde{p}(\mathtt{g} x)\right]$ as the variational ansatz. Since **summation and logarithm do not commute**, it is difficult to compute the model likelihood. Randomly exchanging data, by the Jensen inequality, leads to a bound in the wrong direction for the variational calculation.
>
> To respect the symmetry we performed a summation over the scalar generating function $\varphi(x) =  \frac{1}{|G|}\sum_{\mathtt{g} }\tilde{\varphi}(\mathtt{g} x)$. Sampling the group element in the sum gives an **unbiased estimator** of the model log-likelihood. This is important for physics applications where one wants to claim a rigorous variational upper bound of the free energy.
>
> > ... do you have some guarantees ... the numerical integration has low error? ...
>
> We have confirmed the reported numbers with half integration time-step $\varepsilon=0.05$ and doubled integration steps. The error for a forward-reverse pass is not negligible since the employed Runge-Kutta integration does not respect the time-reversal symmetry. So in both experiments, we only report results for one-way integration.
>
> >  ... Could a time-variant potential function be used?
>
> Using a time-independent parameterization weakens the expressibility. One can certainly employ a time-dependent potential by making the potential  $\varphi(x, t)$ time-dependent. We commented on this extension in Sec. 6 Discussions.
>
> >- Eq (5) is also the Liouville equation ...
>
> Thanks!  Revised.
>
> > ... Figure 3 samples have grey backgrounds ...
>
> That plot is for continuous variables in the $\mathbb{R}^N$ space. Followed Papamakarios et al 2017 we mapped the gray scale MNIST data to **continuous variables**  via $z =\mathrm{logit}(\lambda+(1-2\lambda)x/256)$, where $\lambda=10^{-3}$, and $\mathrm{logit}(\cdot)$ is the inverse of the sigmoid function.
>
> > ... Figure 3 ... is redundant.
>
> We thought that the plot additionally conveys the visual message that training and test losses remain close. We do not have access to the full training curve for other methods.
>
> > Equation references, grammar, and citing format...
>
> Thanks! We fixed these problems.

---

> > ### Comment · AnonReviewer2 · 2018-11-26
> > **Thanks for the clarifications**
> >
> > # "... Randomly exchanging data, by the Jensen inequality, leads to a bound in the wrong direction for the variational calculation."
> >
> > I encourage the authors to make this (and regarding the specific families of "symmetry constraints") more explicit in the text.
> >
> > # "...we mapped the gray scale MNIST data to..."
> >
> > This makes sense, but please also write this explicitly in the text.
> >
> > # "The error for a forward-reverse pass is not negligible..."
> >
> > This could a problem because if the error is not negligible then you may not be solving the differential equations accurately enough. This implies that the log likelihood metrics reported in this paper may not be accurate.
> >
> > ----
> >
> > I have updated my score. Clarity of the paper has improved since the revision, but overall it still has room for improvement and I would have liked to see more experiments. I'm sure there are other problems that motivate the use of symmetry constraints? Also, do the final models actually satisfy these "constraints"? The proposed approach is only a soft regularization approach, so it would've been nice to see for instance the variance of phi(gx) for g sampled from the symmetry group.

---

> > > ### Author Response · Authors · 2018-11-27
> > > **Valueable comments. Will add more revisions and clarifications**
> > >
> > > >  ... encourage the authors to make this (and regarding the specific families of "symmetry constraints") more explicit in the text... "...we mapped the gray scale MNIST data to..." This makes sense, but please also write this explicitly in the text.
> > >
> > > The revision period has passed but we will add these clarifications in our later version of this paper. Thank you for these suggestions, which we believe will make this paper better.
> > >
> > > > "The error for a forward-reverse pass is not negligible..." This could a problem because if the error is not negligible then you may not be solving the differential equations accurately enough. This implies that the log likelihood metrics reported in this paper may not be accurate.
> > >
> > > We have checked the log likelihood metrics reported in the paper using half integration step in the final model. In general, it is indeed a concern to control error. In future work we will consider 1. employ integrators that are reversible by construction (e.g. Leapfrog integration with an implicit start) 2. employ the NeuralODE solver with controlled error https://github.com/rtqichen/torchdiffeq.
> > >
> > > > ... I'm sure there are other problems that motivate the use of symmetry constraints?
> > >
> > > Yes, we believe so. What we are familiar with are examples from physics, chemistry and materials science. For example, besides lattice models, variational free energy calculation for real materials, generation of molecular patterns, solving the Schrodinger equation, etc.
> > > And we envision that imposing symmetries can be beneficial even for density estimation and generative modeling tasks in terms of the inductive bias. In this contexts, we are considering to implement symmetries as hard constraints in the potential network like in https://arxiv.org/abs/1805.09003.
> > >
> > >
> > > > Also, do the final models actually satisfy these "constraints"? The proposed approach is only a soft regularization approach, so it would've been nice to see for instance the variance of phi(gx) for g sampled from the symmetry group.
> > >
> > > Yes. Only the $\mathbb{Z}_2$ symmetry is imposed exactly via the activation function. And the spatial symmetries are imposed in a statistical way. We note one can alternatively implement spatial symmetries exactly in the network architecture. We will add variance in the revised version of the manuscript. Thanks for the suggestions.

---

### Official Review · AnonReviewer1 · 2018-11-02
**Continuous time flows with symmetries motivated from the Monge-Ampere equation**

**Rating:** 7
**Confidence:** 3

**Review:**

Summary:
This paper introduces a continuous-time flow, which is motivated from continuous-time gradient flow in the Monge-Ampere equation in optimal transport theory. They relate the resulting generative model to a dynamical system with a learnable potential function that controls a compressible fluid (representing the probability density), towards the target density. The resulting set of differential equations for the evolution of the samples and the density is solved through a fourth-order Runge-Kutta ODE solver. By imposing symmetries on the scalar potential function, symmetries of the resulting target distribution can also be enforced, a property that is desirable in many applications.

The scalar potential is modeled using a fully connected MLP with a single hidden layer. Forward propagating of samples requires obtaining the gradient of the scalar potential (output of MLP) with respect to its input (the sample). Forward propagation of the log density requires computation of the Laplacian (not the hessian) of the scalar potential. Both of these quantities can easily be computed with automatic differentiation (in O(D) where D is data dimension). The potential is kept constant over time, although this is not necessary.

The proposed method is evaluated on density estimation for MNIST, and variational inference with respect to the Boltzmann distribution of the 2D Ising model at the critical temperature.
On MNIST, comparison is done with respect to MADE, MAF and realNVP. Monge-Ampere flows outperforms the baselines. On the variational inference task one baseline is used, and the result is compared to the exact known free energy. Monge-Ampere flows are reported to approximate the exact solution to comparable accuracy as a baseline. As the authors show that they can easily enforce symmetries, it would be very informative to see the performance of Monge-ampere flows with and without these symmetries enforced on for instance the Ising model. Have the authors looked at this?

It is not clear from the paper how much the ODE solvers used in the forward pass, as well as backpropagating through it with respect to model parameters, will influence the run time. I suspect the training time of models like MAF to be significantly shorter than that of Monge-Ampere flows. For sampling, the comparison would also be interesting. Where sampling from MAF is O(D), with D the data dimension, sampling from the Monge-Ampere flows requires propagating through an ODE solver. Can the authors comment on the runtimes for these settings?

The experimental validation is not extensive, but the proposed method is well motivated and as far as I can tell original. It is a useful contribution to the field of normalizing flows/invertible networks. The ability to easily enforce symmetries into the density seems to be promising and could lead to interesting future work on permutation invariant systems.

See below for comments and more questions:

Quality
The paper is well structured. The experimental validation is not extensive, and perhaps even on the low side.

Clarity
The paper is overal clearly written. One small nuisance is that the citations are not in brackets in sentences, even if they are not part of the actual sentence itself. This interrupts reading. It would be greatly appreciated if the authors could change this. The authors leave out some details with regards to the experiments, but with code available this should be sufficient for reproducibility.

Originality
To my knowledge the idea of using the Monge-Ampere equation for continuous normalizing flows is new. Note that it is also significantly different from a concurrent ICLR submission entitled ‘Scalable Reversible Generative Models with Free-form Continuous Dynamics’, which also discussed continuous normalizing flows with ODE’s.

Significance
This work is of use to the research community. The method is memory efficient and appears to perform well. Especially the ability to enforce symmetries seems very appealing. If the authors can comment on the runtime in comparison to other flow methods, both in terms of training time and sampling, this would enable a better view on practical use.

Detailed questions/comments:

1. In Fig. 3a, the train and test error during training are shown to follow each other very closely. How long was the model trained, and did the train and test curve at some point start to diverge?
2. In Section 4.2, the results are said to be of comparable accuracy as the baseline by Li & Wang. It would be informative to actually state the result of Li & Wang, so that the reader can judge too if this is comparable.
3. Out of curiosity, did the authors also consider using other activation functions that have higher order derivatives, such as tanh?


***** EDIT ******

I thank the authors for their clarifications. They have sufficiently answered my questions/comments, so I will stick with my score.

---

> ### Author Response · Authors · 2018-11-20
> **Very useful and encouraging comments. Addressed all issues, especially symmetry related ones.**
>
> > ... it would be very informative to see the performance of Monge-ampere flows with and without these symmetries enforced on for instance the Ising model. Have the authors looked at this?
>
> Thanks for the suggestions.  Right, we have done this for the Ising model, which motivated us to highlight the importance of imposing symmetries. Please refer to Fig. 4 in our new manuscript. It shows the results for the critical 2D Ising model with and without utilizing the spatial translation (due to periodic boundary conditions)  and $C_4$ rotational symmetries in the variational calculation.
>
> First, the variational free-energy is significantly higher compared to the one with symmetry constraint. Second, by inspecting the generated samples, one sees that the model distribution collapses to the ones with horizontal domains. This phenomenon is related to the “mode collapse” in the training of the generative model. The symmetry breaking solution corresponds to a metastable local minimum of the free energy landscape. By imposing symmetry to the model one performs the variational calculation in the subspace which respects the physical symmetries.
>
> (By the way, the generated samples exhibit two horizontal magnetic domains because we still impose the $\mathbb{Z}_2$ inversion symmetry in the network. Without imposing the inversion symmetry one will generate almost all black/white images due to the ferromagnetic correlations).
>
> In the revised manuscript we added explicitly the discussions made above. We believe that this ablation study, which is also required by Referee 2, should better address the importance of imposing symmetries in physics problems.
>
> > ... I suspect the training time of models like MAF to be significantly shorter than that of Monge-Ampere flows.
>
> Since integrating an ODE is equivalent to a forward pass through a deep residual network, there is no fundamental difference in back propagating through a neural network and an ODE integrator.
>
> As for the absolute training time, we have not compared with MAF. But we noticed that Monge-Ampere flow is significantly slower (10x) than the RealNVP model, which is of a comparable model size. This is because here one has a much deeper network with almost identity transformation in each layer.
>
> > For sampling, the comparison would also be interesting ... Can the authors comment on the runtimes for these settings?
>
> The sampling of the MAF is sequential due to its autoregressive property. While the sampling of the Monge-Ampere flows is parallel over all data dimension, similar to NICE/RealNVP/Glow models.
>
> >  ... Clarity The paper is overal clearly written. One small nuisance is that the citations are not in brackets in sentences ...
>
> We fixed the citations to make it more reader friendly.
>
> > ... Note that it is also significantly different from a concurrent ICLR submission entitled ‘Scalable Reversible Generative Models with Free-form Continuous Dynamics’, which also discussed continuous normalizing flows with ODE’s.
>
> Thank you. We need to acknowledge that our paper is closely related to but significantly different from the FFJORD paper mentioned by the Referee. Despite initially we had a very different motivation and perspective, the final outcome (flow equations 2 and 3) are the same as the ones in the FFJORD paper, and the NeuralODE paper as correctly pointed out by the Referee 2.
>
> >  ... In Fig. 3a ... How long was the model trained, and did the train and test curve at some point start to diverge?
>
> The curve is for 500 epochs with learning rate 0.001. If one continues to train the model one observes that the test curve starts to develop stronger and stronger spikes, which is an indication of overfitting. (One can already observe such feature in the last few epochs of Fig. 3a. )
>
> >  In Section 4.2 ... It would be informative to actually state the result of Li & Wang, so that the reader can judge too if this is comparable.
>
> The relative accuracy of the free energy calculation of  Li & Wang compared to Onsager's exact solution is $0.5\%$. While our variational accuracy is $1\%$.  We mentioned these numbers in the revised manuscript with a remark that Li & Wang has made use of the 2D structure of the problem, while current work does not.
>
> >  Out of curiosity, did the authors also consider using other activation functions that have higher order derivatives, such as tanh?
>
> For now no. We were worried that $\mathrm{tanh}$ might cause vanishing gradient with large pre-activation. In our implementation, we were using $\mathrm{softplus}(x)$ [and $\ln\cosh(x)$ for cases one wants to enforce inversion symmetry]. We notice that the activation functions that we used here should also have higher order derivatives.

---

### Public Comment · (anonymous) · 2018-11-06
**A Related Paper by Mesa, et al**

A similar idea of using the optimal transport for MNIST generation has been considered in this paper:
https://arxiv.org/pdf/1801.08454.pdf

The author(s) need to address this.

---

> ### Author Response · Authors · 2018-11-20
> **An interesting paper. Cited.**
>
> Thanks for pointing this out this interesting paper to us.
> It appears the nonequilibrium thermodynamics approach involving Langevin equation is more in line with Sohl-Dickstein 2015. Addressing the general framework of optimal transport for generative modeling is nevertheless similar to our manuscript.
> We cited this paper in the revised manuscript. Thank you!

---

### Public Comment · ~Anirudh_Goyal1 · 2018-11-28
**More related work.**

I enjoyed reading the paper. The idea of combining the forward KL as well as minimizing the reverse KL (without the need to sample) is an interesting one. : )

We have also done some work  which allows to train generative models, as by training a transition operator of a markov chain by directly parameterizing the transition operator using a function approximator. We get rid of sampling process, by sharing the transition operator for inference as well as generative phase (though, we do need to increase temperature etc)

1) Variational Walkback (https://arxiv.org/abs/1711.02282)

Thanks for your time! :)

---

> ### Author Response · Authors · 2018-12-03
> **Interesting work!**
>
> This is indeed a very interesting idea. Thank you for sharing this work with us!
> We will address it in our revised version!

---

### Comment · Area_Chair1 · 2018-12-12
**Worried about numerical error biasing likelihood scores.**

I'm concerned about the numerical error introduced by the approximate, fixed-step integrator used.  In the paper, the authors did not check the degree of numerical error (or to what extend their reported likelihoods do not normalize) as a function of the step size.  In the comments below, you state: "We have checked the log likelihood metrics reported in the paper using half integration step in the final model. In general, it is indeed a concern to control error."

In general, because the dynamics of the forward integrator are differentiated during training, we should expect the reported likelihoods to be systematically over-estimated in this methods' favor, as the optimization learns to produce numerical error in the direction that helps its objective.

Can you tell us what you found when you checked the numerical error?  Are there any reassurances you can give that the likelihoods you report normalize to 1?

---

> ### Author Response · Authors · 2018-12-13
> **Thanks! Numberical error checked and consistent results observed**
>
> > I'm concerned about the numerical error introduced by the approximate, fixed-step integrator used.  In the paper, the authors did not check the degree of numerical error (or to what extend their reported likelihoods do not normalize) as a function of the step size.  In the comments below, you state: "We have checked the log likelihood metrics reported in the paper using half integration step in the final model. In general, it is indeed a concern to control error."
> > In general, because the dynamics of the forward integrator are differentiated during training, we should expect the reported likelihoods to be systematically over-estimated in this methods' favor, as the optimization learns to produce numerical error in the direction that helps its objective.
> > Can you tell us what you found when you checked the numerical error?  Are there any reassurances you can give that the likelihoods you report normalize to 1?
>
> Thanks for your comment! We agree that the numerical error should be more carefully checked. Here is a summary of our numerical check for density estimation
>
> | Integration steps |  Test NLL  |
> | ------------------------- | ---- |
> | $\varepsilon=0.1, d=100$ | 1254.045 |
> | $\varepsilon=0.05, d=200$ | 1254.627 |
> | $\varepsilon=0.02, d=500$ |1255.625|
> | $\varepsilon=0.01, d=1000$ |1255.236|
> |                           |      |
>
> These checks can be reproduced by loading the pretrained model we posted on Github and adjusting parameters `epsilon` and `Nsteps` respectively. The deviation of the mean test NLL is smaller than the statistical error bars reported in the manuscript ($1255.5\pm 2.0$). Based on these observations, we believe the value reported in the manuscript is meaningful.
>
> It is exactly for the purpose of controlling the integration error we had chosen the fourth order Runge-Kutta integration scheme which has $O(\varepsilon^4)$ accumulated error. We note that for small scale problems it is possible to check the normalization by using Reimann sum on a very fine grid, c.f. https://arxiv.org/abs/1810.01367. However, it is difficult to check normalization of the likelihood for high dimensional distribution.
>
> An (indirect) reassurances about the normalization is that our variational calculation of the Ising model reaches $1\%$  accuracy to its exact solution. If there was a problem with the normalization one would violate the variational principle and obtain even lower variational free energy.
>
> Thank you!
>
> P.S. We report additional numerical checks for the variational free energy calculation of the Ising model for the record. The error bars correspond to one standard deviation evaluated on batch size of $64$.
>
> | Integration steps          | Variational free energy |
> | -------------------------- | ----------------------- |
> | $\varepsilon=0.1, d=100$   | -2.2951 $\pm$ 0.0016    |
> | $\varepsilon=0.05, d=200$  | -2.2960$\pm$ 0.0015     |
> | $\varepsilon=0.02, d=500$  | -2.2966 $\pm$ 0.0014    |
> | $\varepsilon=0.01, d=1000$ | -2.2969 $\pm$ 0.0014    |
> |                            |                         |
>
> We notice that for this problem there are two kinds of errors: systematical error (e.g., due to numerical discretization) and statistical error. Here the reported accuracy of the variational free energy is valid with these errors considered.

---

### Meta-Review · Area_Chair1 · 2018-12-11
**Interesting ideas but has fundamental technical problems**

**Confidence:** 2
**Recommendation:** Reject

**Metareview:**

This paper develops a generative density model based on continuous-time flows on a potential field.

Strengths:  The paper contains interesting ideas and connections to physics, in particular the enforcement of symmetry in a computationally cheap way.

Weaknesses:  The main quantitative results of this paper are undercut by the numerical error introduced by the approximate, fixed-step integrator used.  In the paper, the authors did not check the degree of numerical error (or to what extend their reported likelihoods do not normalize) as a function of the step size.  This was partially addressed in a comment below.

There does seem to be some novelty but the lack of concrete experiments is a letdown. One could e.g. verify that the samples have similar properties (e.g. moments) to the ground truth, which are known for the Ising model. Regarding clarity, the symmetry constraints are never clearly specified.

This paper contains many ideas that would have been novel, but were scooped by [1] which was put on arXiv 3 months before the ICLR submission date.  The authors have added appropriate references to this paper, but this still undercuts the originality of the contribution.

The explanation of how and which symmetries are enforced is a little bit buried and unclear.

Points of contention:  Two of the reviewers didn't seem to be aware that the main mathematical results of the model are special cases of results from [1].

Consensus:  All reviewers agreed that there were interesting ideas in the paper, and that it was close to the bar.

[1] Chen, Tian Qi, et al. "Neural Ordinary Differential Equations."